
# Catchment landforms predict groundwater-dependent wetland sensitivity to recharge changes

Etienne Marti[1], Sarah Leray[1,2], Clément Roques[3]

[1] Departamento de Ingeniería Hidráulica y Ambiental, Pontificia Universidad Católica de Chile, Santiago, Chile
[2] Centro de Cambio Global UC, Santiago, Chile
[3] Centre for Hydrogeology and Geothermics (CHYN), University of Neuchâtel, Neuchâtel, Switzerland

*Correspondence to*: Clément Roques (clement.roques@unine.ch)

**Abstract.** This study investigates the influence of topography on the desaturation rates of groundwater-dependent wetlands in response to changes in recharge. We examined sixty catchments across northern Chile, which feature a wide variety of landforms. We categorized the landforms using geomorphon descriptors, identifying three distinct clusters: lowland, transition, and mountain settings. Using steady-state 3D groundwater models, we derived flow partitioning and seepage area extent for each catchment. Each cluster revealed consistent seepage areas evolution under varying wet-to-dry conditions. Our findings indicate that mountains exhibit reduced seepage area compared to lowlands at equivalent hydraulic conductivity to recharge (K/R) ratios but are less sensitive to recharge fluctuations with slower rates of seepage area variation. Statistical evidence demonstrates that geomorphons-defined landforms correlate with desaturation indicators, enabling the prediction of catchment sensitivity to climate change based solely on a topographic analysis.

**Short summary.** The research demonstrates that the response of groundwater-dependent wetlands to recharge changes can be accurately predicted solely based on landform properties, providing a practical and scalable approach for wetland vulnerability assessment. We reveal that mountain catchments are less sensitive to recharge changes than lowland catchments - due to fewer but more persistent seepage areas. It offers critical insights for evaluating the vulnerability of catchments to climate change impacts and has direct implications for water resource management and conservation planning in diverse landscapes.

## 1. Introduction

Groundwater seepage occurs when the water table intersects the topography. Thus, landforms influence both its spatial distribution and temporal dynamics (Bresciani et al., 2014, 2016; Sophocleous, 2002). The interaction between groundwater and topography significantly impacts the resilience of groundwater-dependent wetlands to climate variability (Cuthbert et al., 2019; Scanlon et al., 2023). Considering steady-state groundwater flow systems, the depth of the water table, and so the distribution of flow paths and groundwater seepage areas, are controlled by the recharge rate ($R$), the topography and by the hydrodynamic properties of the aquifer through its hydraulic conductivity ($K$), (Condon & Maxwell, 2015; Rath et al., 2023; Tóth, 1963; Zhang et al., 2022). An equivalence of effects between $R$ and $K$ has been demonstrated (Bresciani et al., 2014;



Haitjema & Mitchell-Bruker, 2005; Jamieson & Freeze, 1982), allowing a convenient focus on the dimensionless $\frac{K}{R}$ ratio and the topography. The groundwater table is typically near the surface in low-relief and/or humid regions, and deeper in rugged terrain and/or arid regions. However, the hydrogeological response and seepage dynamics to varying landscapes and topographic features are not straightforward and difficult to predict.

Analytical solutions have been proposed to quantify the extent of groundwater seepage under varying $\frac{K}{R}$ at the hillslope scale, using simplified groundwater flow equations (Bresciani et al., 2014, 2016). Marçais et al. (2017) conducted modeling experiments to estimate seepage extent and dynamics using a 2D representation of the equivalent hillslope. While these approaches are applicable in shallow aquifers, where flow predominantly follows the topography, they do not capture the complexity of 3D groundwater flow, especially under low water tables or steep reliefs. A few 3D numerical modeling

experiments have been undertaken, mainly for sensitivity studies with conceptual surface and subsurface geometries (Carlier et al., 2019; Gauvain et al., 2021; Gleeson & Manning, 2008; Welch et al., 2012). There is a pressing need to understand seepage distribution and dynamics, considering the intricate topographic nuances of real-world conditions and the 3D nature of groundwater flow. This knowledge is essential to predict the extent of groundwater seepage and its dependent wetlands under future climate scenarios.

Our study aims to model the partitioning of 3D groundwater flows and their seepage extent across different landscapes, from lowland to high mountains, considering various $\frac{K}{R}$ values. Additionally, we aim to identify appropriate topographic indicators that explain the variety of hydrogeological behaviors, providing statistical means to extrapolate our results to other contexts. We developed a parsimonious 3D groundwater flow modeling approach which we applied to sixty catchments along 1,800 km of the northern Chile. This choice was motivated by the rich diversity of geomorphological contexts allowing the exploration

of a wide range of hydrogeological responses.

## 2. Material and Methods

### 2.1. Geomorphological context

The study area is located in northern Chile between Santiago and the Peruvian border (~1,800 km long). The landforms diversity results from the specific tectonic and weathering processes involved in the Andes. This process resulted in the

formation of a longitudinal valley called the Central Depression, which is bounded by two cordilleras, the Coastal Cordillera and the Principal Cordillera, both composed mainly of volcanic-sedimentary rocks (Hartley & Evenstar, 2010; Jordan et al., 1983). The Coastal Cordillera forms an intermediate mountain range with an average elevation between 1,000 and 2,000 m.a.s.l., while the Principal Andean Cordillera has maximum elevations close to 7,000 m.a.s.l. (Figure 1a) (Armijo et al., 2015; Charrier et al., 2007). The Central Depression delineates a sedimentary basin rich in Quaternary alluvial deposits with variable

thicknesses, ranging from about 250 m near Santiago (Yáñez et al., 2015) to almost 1,000 m in the northern regions of Chile





(Hartley & Evenstar, 2010; Jordan et al., 2014; Nester & Jordan, 2011). Specifically, the Pre-Cordillera is a transition zone between the Central Depression and the Western Cordillera, with catchment characteristics that vary from nearly flat terrain to mountainous regions with steep slope gradients and typical mountain front geomorphology (Figure 1a). The diversity of these environments provides an excellent opportunity to explore a wide range of topographic settings, including flat catchments, mountain-front areas, incised mountain catchments, volcanoes, and high mountain peaks.





**Figure 1: Methodological workflow for topographical classification and analysis. (a) Location and topographic map of the study area in northern Chile, with the boundaries of the 60 studied catchments highlighted in black. (b) Example of landform classification within a single catchment, including a landform map, the proportion of landform types, and an illustration of the principal landform**
**categories defined by geomorphons (adapted from Jasiewicz & Stepinski, 2013). (c) Catchment categorization using Principal Component Analysis (PCA) for dimensionality reduction and clustering. The selected catchment is plotted in the PCA space, illustrating its position relative to the first two principal components (PC1 and PC2). Red arrows represent the eigenvectors associated with different landform types, showing their contribution to the PCA axes.**

### 2.2. Classification of catchment topographies

We considered the catchment boundaries of the global catchment database HydroATLAS (Lehner & Grill, 2013). We chose to limit the size of the catchments to between 500 and 1,500 km$^2$ by considering level 8 of HydroATLAS. To cover different geomorphological and tectonic settings, we selected sixty catchments within our study area (indicated by colored boundaries in Figure 1a). We extracted the topography from the digital elevation model (DEM) from the SRTM (Shuttle Radar Topography Mission, 90 m resolution).

We used the geomorphons classification method proposed by Jasiewicz & Stepinski (2013) to categorize the topographies into ten different landforms (Figure 1b). Geomorphons-defined landforms are italicized throughout the text for improved readability. This methodology is based on elevation differences in eight directions relative to the reference cell. This operation is reproduced for each cell of the DEM, identifying a shape for each of these cells (Figure 1b). Unlike the direct cell neighbor method (e.g. slope, curvature, or roughness), the geomorphons method allows to capture landforms at larger scales by defining
a search radius around the reference cell, the look-up distance in Jasiewicz & Stepinski (2013). Here we defined it as a function of the hillslope characteristic length:

$$L = \frac{1}{2D} \rightarrow L = \frac{1}{2*\frac{l}{A}}, \qquad\qquad \textbf{Equation 1}$$

where $L$ is the catchment feature length, $l$ is the river network length, $D$ is the drainage density and $A$ is the catchment area. The river network is defined using a surface flow accumulation routine available in the Whitebox tool Python package (Lindsay, 2016). Combining principal component analysis (PCA) and k-means clustering (Figure 1c), we categorized the
catchments by their dominant topographical features based on landform proportions (Figure 1b). PCA is a classical statistical method used to reduce the dataset dimensionality by transforming the original variables into a new set of uncorrelated variables, called principal components (PC), which capture the maximum variance in the data. The k-means clustering approach allows us to identify groups of catchments belonging to the cluster with the nearest mean within the new PC space (Figure 1b and c). Python code and trained models are available on repository (Marti et al., 2024).

### 2.3. Numerical modelling of seepage area

A three-dimensional numerical groundwater flow model was developed for each catchment. The models were constructed and run using the MODFLOW-2005 software suite (Harbaugh, 2005) with the NWT solver (Niswonger et al., 2011), and managed





through the Python-based interface FLOPY (Bakker et al., 2016). The diffusivity equation was solved under steady-state conditions for unconfined flow.

The horizontal discretization followed the DEM resolution, set at 90 meters (Figure 2a), while the vertical discretization consisted of ten layers with exponentially increasing thickness (Figure 2b). A buffer zone around each catchment expanded the modeled domain area by 20%, ensuring boundary conditions did not impact seepage distribution within the studied catchment (Figure 2a). The model bottom mirrored the topography with a 100m-thick aquifer (Figure 2b). Assuming a constant aquifer thickness minimized the potential effects of transmissivity changes on seepage distribution. The 100m thickness was

chosen to realistically accommodate both flat sedimentary catchments and steep mountainous aquifers (Condon et al., 2020). The side and bottom boundaries of the buffer box were set as no-flow. For generality, effective recharge $R$ was uniformly set at the water table, and a drain was set on the topography using the eponymous packages in MODFLOW. Hydraulic conductivity ($K$) was set to be homogeneous and isotropic.

Various water table positions relative to the topography were derived by setting different values of the $\frac{K}{R}$ ratio, ranging from

100 for fully saturated conditions to 10,000 when all simulated catchments reached near full desaturation $\frac{seepage\ area}{catchment\ area}$ ($S_A^*$) < 1%. $\frac{K}{R}$ values were logarithmically spaced within this interval, and simulations were stopped if the catchment's seepage area fell below the 1% threshold (Figure 2c). This modeling workflow resulted in a total of 1,793 simulations.

For each catchment, we perform a power law fit on the relationship between seepage area extent and $\frac{K}{R}$ further mentioned as

the desaturation function (Equation 2a, and red curve in Figure 2c). This allows us to capitalize on the observed linear relation between log ($S_A^*$) and log ($\frac{K}{R}$):

$$S_A^* = \left(1 + \left(\frac{\frac{K}{R}}{\lambda}\right)^2\right)^n .$$

**Equation 2a**

$$\frac{dS_A^*}{d\frac{K}{R}} = \frac{2n}{\lambda^2} \frac{K}{R} \frac{S_A^*}{1 + \left(\frac{\frac{K}{R}}{\lambda}\right)^2}$$

**Equation 2b**





$$\frac{dS_A^*}{d\frac{K}{R}} \approx \frac{2nS_A^*}{\frac{K}{R}} \ when \ \frac{K}{R} \gg \lambda$$

**Equation 2c**

The desaturation function is determined by the proportionality constant, $\lambda$, which can be associated with a desaturation threshold, i.e. the critical value of $\frac{K}{R}$ above which the catchment begins to desaturate. The negative desaturation exponent, $n$, directly affects the rate of change in seepage extent as $\frac{K}{R}$ increases, as shown in Equations 2b and 2c. It can be viewed as a

measure of the sensitivity of the seepage area extent to a deepening of the water table: for a given pair of seepage area and $\frac{K}{R}$, a lower $n$ indicates a higher sensitivity of the catchment to a decrease of the water level. We estimated $n$ considering seepage area extents lower than 20% of the catchment area, which are more representative of real-world conditions i.e. by giving more weight in the fit to the higher $\frac{K}{R}$ ratios.

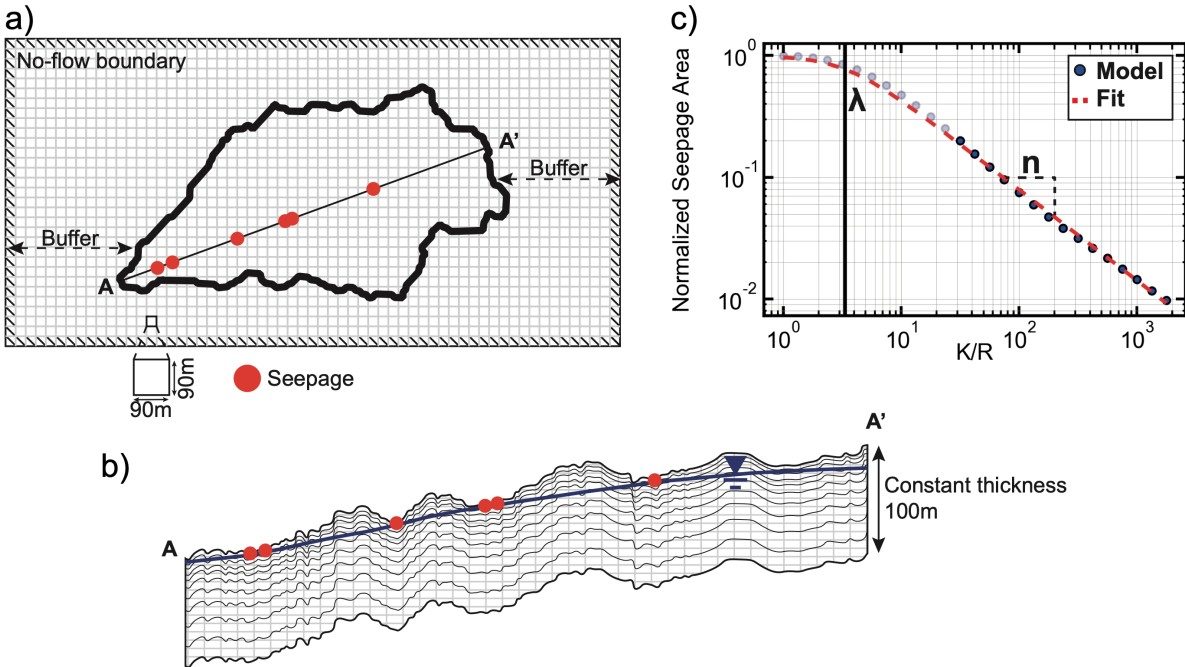

**Figure 2: Model settings for one example catchment: (a) Horizontal discretization and buffer zone beyond catchment limits, (b) AA' cross-section including vertical discretization and an arbitrary water table intercepting the topography creating seepage areas (red dots), (c) Example of a results of normalized seepage area with respect to K/R (blue dots) with the power law fit for Eq. 2a (dashed red curve).**



## 3. Results

### 3.1. Typology of catchments topography

We evaluate the proportion of main geomorphons on the sixty-catchment dataset. The PCA analysis (Figure 1c) resulted in a first component (PC1) explaining 75.4% of the total variance, and the first two components (PC1 and PC2) combined explaining 86.8% of the total variance. This result indicates a strong relation between the landforms description from geomorphons analysis in the catchments and the reduction into 2 dimensions. We found that PC1 mostly represents the differentiation between *flat* and *slope* landforms, with eigenvector magnitudes of -0.4 for the *flat* landform and 0.4 for the *slope* landform. *Slope* landform is associated with the *peak*, *ridge*, *valley*, *hollow*, *spur*, and *pit* forms, all showing similar eigenvectors magnitude on PC1. Regarding PC2 the *footslope* and *shoulder* forms are the main control with an eigenvector magnitude of respectively, 0.8 and 0.5. *Flat* and *slope* forms show eigenvectors magnitudes of -0.2 and 0.2 along the PC2 axis. To further support and illustrate this description, the catchments are grouped into three clusters (colored dots in Figure 1c and colored catchments contour on Figure 1a). The red cluster is highly influenced by the *flat* landform as lowland catchments would. Conversely, the blue cluster is strongly influenced by *slope* and associated landforms along PC1 (*ridge*, *valley*, *peak*) representative of mountain catchments with incised valleys, and narrow valley bottoms. The green cluster exhibits more dispersion in the represented catchments. This cluster acts as a transition between the red and blue clusters. It contains catchments most influenced by *footslope* and *shoulder* forms. Positioned centrally between the two extreme clusters, this cluster serves as a transition zone between flatter areas and mountain catchments, possibly including catchments showing both flat and mountainous characteristics, as observed at mountain fronts.

To facilitate comprehension and illustration, these clusters are referred to as *lowland cluster* for the red cluster, *mountain cluster* for the blue cluster and *transition cluster* for the green cluster thereafter.

### 3.2. Seepage distribution evolution with increasing $\frac{K}{R}$

Figure 3a illustrates the evolution of seepage area, normalized by catchment area, as a function of $\frac{K}{R}$ for all sixty catchments. Four specific catchments are highlighted with sharper lines for further discussion. As expected, lower $\frac{K}{R}$ values result in fully saturated catchments, while as $\frac{K}{R}$ increases, all catchments progressively desaturate at varying rates. For instance, at a normalized seepage area of 20%, the corresponding $\frac{K}{R}$ values range from 20 to 250.

Similar to the distinct landform clusters, the desaturation behavior of these clusters is clearly defined (Figure 1), confirming that variations in seepage distribution are predominantly driven by topographical effects. The power law fit of seepage distribution (Equation 3a) for each of the sixty catchments results in $\lambda$ values ranging from 2.05 to 37.03 and n values ranging



from -0.44 to -0.31. The fit shows minimal RMSE values between 0.01 and 0.08, indicating that seepage evolution with increasing $\frac{K}{R}$ can be successfully parameterized with only two parameters, $\lambda$ and $n$.

Regarding the desaturation threshold, $\lambda$, the mountain cluster shows lower values than the lowland cluster. The transition
160 cluster demonstrates intermediate behavior, reaching higher $\lambda$ values than the mountain cluster but lower than the lowland cluster. Regarding the desaturation exponent, $n$, within the low-saturation domain ($\leq 20\%$), the mountain cluster exhibits slower desaturation rates, while the lowland cluster presents faster desaturation rates. The variability in desaturation slopes for the transition cluster is more pronounced, reflecting a mix of behaviors within this zone, yet it again shows intermediate behavior relative to the other two clusters.



Figure 3: (a) Normalized seepage area against $\frac{K}{R}$ for the sixty catchments (colored lines with markers), including log-log plot on the upper-right corner. Four illustrative catchments are highlighted: mountain type M1 (blue diamond, (b)), lowland type L2 (red cross,

165





(c)), transitory type T3 (green triangle, (d)) and T4 (green cross, (e)). Details include the topographic map overlayed by the seepage area at an arbitrary value of 20% (red mask) with corresponding $\frac{K}{R}$ value followed by the landforms histogram.

### 3.3. Seepage patterns in representative catchments

Figure 3 highlights four distinct catchments: M1, representative of the mountain cluster; L2, representing the lowland cluster; and T3 and T4, showing the range of responses within the transitory cluster. We present the seepage distribution over the topographic map and the landform proportions for each catchment (Figure 3b, c, d, and e).

For M1 (Figure 3b), characterized by low $\lambda$ and high $n$ values, it exhibits the typical seepage distribution of mountainous regions. At a normalized area of 0.2, seepage primarily congregates in topographic lows, such as river valleys, while ridges and peaks desaturate due to their significant elevation compared to the surrounding terrain. Conversely, L2 (Figure 3c), with high $\lambda$ and low $n$ values, suggests that the water table remains closer to the surface in lowland settings.

For T3 and T4, their landform proportions (Figure 3d and e) reveal similar values for most forms, except for a higher proportion of *shoulder* and *footslope* forms in T4. This increased prevalence of *shoulder* landform in T4 is due to a prominent incised river valley in the eastern part of the catchment.

Examining T3's seepage distribution, it initially aligns with the mountain cluster with a low desaturation threshold ($\lambda$). Then, in the range of $1 < \frac{K}{R} < 10$, a substantial change in slope is observed, with the distribution intersecting that of the lowland cluster for high $\frac{K}{R}$ values, ultimately being the last catchment to reach a saturation level of <1%. This behavior can be explained by looking at the spatial distribution of seepage for T3 (Figure 3d). A clear demarcation exists between the flat western area and the mountainous settings to the east. At higher elevations, desaturation occurs at lower $\frac{K}{R}$ values, resulting in a low desaturation threshold ($\lambda$). Subsequently, at a normalized area of 0.2, the catchment behaves like the lowland cluster, influenced by the western part of the catchment.

Conversely, T4's seepage distribution exhibits an opposite pattern. It initially mirrors the lowland cluster with a higher $\lambda$ value, ultimately resembling the mountain cluster characteristics, reaching saturation levels under 1% for similar $\frac{K}{R}$ values. The seepage spatial distribution for T4 (Figure 3e) shows that higher elevation zones in the western part of the catchment have already undergone desaturation and tend to develop exclusively within topographic lows, specifically at the bottom of the singular river channel.

### 3.4. Linking topographic features and desaturation behavior

We computed the correlation matrix between the principal components (PC1 and PC2) and the desaturation parameters ($\lambda$ and $n$) to assess the strength of the topographical control on desaturation behavior. Figure 4a shows a strong anti-correlation between $\lambda$ and PC1, with a Spearman coefficient of r = -0.96 (p < 0.0001). Figure 4b displays a strong linear correlation





between $n$ and PC1, with r = 0.76 (p < 0.0001). No other significant correlations were identified (the entire correlation matrix is available in Supplementary Material S1).



Figure 4: Scatter plots for the original sixty-catchment dataset (dots) with Spearman coefficient r, the equation and the coefficient of determination (R²) of a global fit (black line with 95% confidence interval) for (a) λ against PC1 and for (b) $n$ against PC1. Random Forest predictions for both parameters are overlayed on the original data (crosses). (c) PCA plot, for the original (dots) and the prediction (crosses) datasets. The percentage of variance in the original dataset explained by each component is displaid on



**axis title. (d) and (e) Situation and topographic maps of the study area highlighting the one hundred and twenty-three catchments**
**with a colored according to the defined clusters. Each catchment is overlayed by a dot (original dalaset) or a triangle (prediction dataset) which size depends on $\lambda$ and the color for $n$.**

The clusters are well distinguishable in Figure 4a and 4b. In Figure 4a, the inverse relationship between $\lambda$ and PC1 is robustly quantified by fitting an exponential function ($R^2 = 0.89$), facilitating a quantitative correlation between $\lambda$ and PC1, as illustrated in Figure 4a. The mountain cluster is isolated with a low average and variance in $\lambda$ values. The transition cluster forms the
elbow part of the exponential decay, while the lowland cluster is clearly identified with higher $\lambda$ values. The variations in $\lambda$ values are higher in the lowland and transition clusters compared to the mountain cluster.

Figure 4b reveals a linear relationship between $n$ and PC1. The excellent linear fit ($R^2 = 0.72$) allows for a straightforward quantification of the relationship between PC1 and the negative scaling exponent $n$ and consequently the appraisal of the desaturation rate based on landforms (Equation 2b and c). The clusters are well identified, with the lowland cluster showing
lower $n$ values than the mountain cluster.

We finally employed a Random Forest regression on the dataset to predict $\lambda$ and $n$ based on topographic parameters (PC1 and PC2) for sixty-three catchments located both in the same study area and expanding further South inside Chile (Figure 4d and e). We defined PC1, PC2 and clusters for these new catchments using the originally trained PCA and k-means model (Figure 4c). Regarding, training and testing of the Random Forest algorithm, we used the original sixty catchments dataset. This initial
analysis involved 5,000 iterations of sampling with replacement, each using 10 test catchments, with the remaining 50 catchments used for training. This resampling approach was adopted to assess the robustness of the estimations in the presence of random variations within the selected test and training data and was evaluated calculating the coefficient of determination ($R^2$) within the tested data (see Supplementary Material S2 for Kernel Density Estimate (KDE) plot of $R^2$ distribution). We defined the model used for predictions based on the best compromise to estimate both $\lambda$ and $n$. Hyperparameters (number of
trees and maximum depth) were tuned using cross-validation techniques. Predictions made for $\lambda$ (Figure 4a) show good consistency with the original dataset both in terms of identifying clusters behaviors and in trend, following the originally defined exponential relation. Regarding $n$ (Figure 4b), we observe a similar accuracy to represent clusters, while the general linear trend is less obvious. We observe for $n$, while following an increasing trend, diversified response between the clusters with the *transition cluster* exhibiting a greater rate of change in $n$ for an equivalent increment in PC1. Yet it is a better fit to
original data. The spatial distribution of the predicted catchments observed on Figure 4d and e, is a good match with the original data both in terms of topographic characteristics and desaturation function parameters.

## 4. Discussion and perspectives

Groundwater flow and storage regulate the resilience of wetlands to climate changes (Fan et al., 2019). Variations in topography and landforms across catchments lead to differences in wetland sensitivity to changing recharge by shaping distinct
groundwater flow structures. In this study, we provide a quantitative assessment of the controls of landforms on the sensitivity



of groundwater-dependent wetlands to aquifer desaturation (through variations in $\frac{K}{R}$) across sixty catchments along Nothern Chile, covering settings from lowlands to mountains. These feedback mechanisms were analyzed using a novel combination of three-dimensional process-based groundwater modeling, geomorphons-based landforms characterization, and multivariate statistical analysis at the regional scale.

Our results demonstrate that the desaturation functions of catchments can be explained by the typologies in topographies derived from landform categorization. Building on previous works that focused on two-dimensional aquifer geometry, as first introduced by Haitjema and Mitchell-Brucker (2005) and further explored by Bresciani et al. (2014), we show that mountainous regions exhibit lower seepage extents, restricted to incised valleys, compared to lowland catchments at equivalent $\frac{K}{R}$ ratios. However, we demonstrate that mountainous regions are less sensitive to changes in saturation, exhibiting slower desaturation

rates.

To disentangle the respective impacts of different landforms, we compared our results with those obtained from the analytical solution proposed by Bresciani et al. (2014) for a 1D hillslope, where one can easily assess the impacts of slope angle and the concavity/convexity of the hillslope (results in Supplementary Material S3). In agreement with our results, the steepness of the hillslope is the primary influence on seepage extent and its variation through changes in groundwater level. Steep slopes begin

to desaturate at lower $\frac{K}{R}$ values than gentler slopes. Additionally, for a given change in groundwater level, the rate of change in seepage extent is inversely correlated with slope angle. This aligns with the differences in the desaturation exponent, $n$, and the desaturation threshold, $\lambda$, obtained for the mountain and lowland clusters. Mountain clusters have higher $n$ and $\lambda$ values, suggesting higher resilience to changes in $\frac{K}{R}$.

Furthermore, the analysis of the simple analytical solution demonstrates that hillslope shape (concave vs. convex) also affects

the desaturation function, though to a lesser extent than slope. Concave slopes appear to have a lower $\lambda$ but a higher $n$ than convex slopes. Similarities between concave and convex hillslopes can be found in the *shoulder* vs. *footslope* in our landform classification. *Shoulder* and *footslope* is differentiated primarily along PC2, explaining a smaller proportion of the variance in the dataset analyzed here. However, no clear correlation between PC2, $n$, and $\lambda$ was found, suggesting a minimal impact of *shoulder* and *footslope landforms* compared to the other ones.

Our results establish a robust statistical framework demonstrating a strong correlation between landforms, categorized by the dominant landforms (PC1), and hydrological parameters that assess the sensitivity of groundwater seepage to desaturation with changing recharge. This framework allows for predictions using straightforward statistical learning techniques. The Random Forest algorithm yields highly promising results for the sixty-three catchments estimated. This approach provides valuable insights into assessing catchment vulnerability to climate change on a regional scale, even for ungauged basins (Hrachowitz

et al., 2013).



## Acknowledgments

We acknowledge funding from the Agencia Nacional de Investigación y Desarrollo (ANID) through grants Fondecyt Regular
n°1210221, Anillo n°ATE220055, and Anillo n°ATE230006.

## Open Research

The HydroATLAS global catchments database is freely available at https://www.hydrosheds.org/hydroatlas. The SRTM
Digital Elevation model is available on NASA website (https://www.earthdata.nasa.gov/sensors/srtm). Python code to
reproduce the analysis, the trained models (PCA, k-means and Random Forest), and the data used to generate results and
figures in this manuscript is publicly available at Marti et al (2024). This includes landfoms proportion, PCA, cluster
information, $\lambda$ and $n$ values, and seepage area distribution for each catchment of the original dataset. The predicted dataset is
also included (PCA, clusters, $\lambda$ and $n$ estimations). Catchments ID correspond to the HydroATLAS identification.

## Competing interests

The contact author has declared that none of the authors has any competing interests.

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
