# Peer review of "Catchment landforms predict groundwater-dependent wetland sensitivity to recharge changes"

_Hydrology and Earth System Sciences, 2024_

## Referee Comment (RC2)

5

10

20

**Catchment landforms predict groundwater-dependent wetland sensitivity to recharge changes**

Etienne Marti1, Sarah Leray1,2, Clément Roques3

1 Departamento de Ingeniería Hidráulica y Ambiental, Pontificia Universidad Católica de Chile, Santiago, Chile

2 Centro de Cambio Global UC, Santiago, Chile

3Centre for Hydrogeology and Geothermics (CHYN), University of Neuchâtel, Neuchâtel, Switzerland

Correspondence to: Clément Roques (clement.roques@unine.ch)

Abstract. This study investigates the influence of topography on the desaturation rates of groundwater-dependent wetlands in response to changes in recharge. We examined sixty catchments across northern Chile, which feature a wide variety of landforms. We categorized the landforms using geomorphon descriptors, identifying three distinct clusters: lowland, transition,

- and mountain settings. Using steady-state 3D groundwater models, we derived flow partitioning and seepage area extent for each catchment. Each cluster revealed consistent seepage areas evolution under varying wet-to-dry conditions. Our findings indicate that mountains exhibit reduced seepage area compared to lowlands at equivalent hydraulic conductivity to recharge (K/R) ratios but are less sensitive to recharge fluctuations with slower rates of seepage area variation. Statistical evidence
- 15 demonstrates that geomorphons-defined landforms correlate with desaturation indicators, enabling the prediction of catchment sensitivity to climate change based solely on a topographic analysis.

**Short summary.** The research demonstrates that the response of groundwater-dependent wetlands to recharge changes can be accurately predicted solely based on landform properties, providing a practical and scalable approach for wetland vulnerability assessment. We reveal that mountain catchments are less sensitive to recharge changes than lowland catchments - due to fewer but more persistent seepage areas. It offers critical insights for evaluating the vulnerability of catchments to climate change impacts and has direct implications for water resource management and conservation planning in diverse landscapes.

**1. Introduction**

[revised manuscript text omitted]

---

## Author Comment (AC1)

**Our answers in *Italic*.**

**Reviewer #1 Evaluations:**

Thank you for submitting your Manuscript to HESS. Please find below the comments and suggestions made regarding the current version of the manuscript.

This manuscript aims to study the influence of topography on the desaturation rates of groundwaterdependent landscapes in response to changes in recharge. The authors achieve this objective by categorizing the catchments into lowland, transition, and mountain settings clusters, using geomorphon descriptors, and implementing 3D steady-state groundwater models to derive each catchment's flow partitioning and seepage area extent. The findings illustrate that mountains exhibit reduced seepage area compared to lowlands at equivalent hydraulic conductivity and recharge ratios but are less sensitive to fluctuations in recharge. Finally, the authors performed a correlation statistical analysis between the geomorphon-define landforms and the desaturation indicators, which, according to them, enables the prediction of sensitivity to climate change based on topographic analysis.

I have read the manuscript with great interest. My overall opinion is that the manuscript is wellwritten but needs some moderate revisions and clarifications. Below, I have listed comments, hoping they may help improve the manuscript's quality.

We would like to thank the reviewer for the interest shown in our work and for the constructive evaluation. We appreciate the recognition of our modelling approach and the valuable suggestions provided to improve clarity. All comments will be carefully addressed and incorporated into the revised manuscript.

**Specific Comments**

- 1. Some clarification is needed regarding the 3D numerical model approach.
  - The authors extended the model domain by 20% of the total area of the catchment to avoid boundary effects. In the extended domain, did the authors use the elevation of adjacent catchments? Have the authors considered possible interbasin flow happening among catchments, as described in Fan (2019)? If so, are there any implications for this interbasin flow in the general results for lowland settings?

Indeed, the elevation of the adjacent catchments was used. Recharge was applied on all model cells, including the buffer area around the catchment, while the catchment is in the center of the simulated area. The 60 modeled catchments and their boundaries were validated by analyzing the topographic context in the area (topographic maps of each catchment and its buffer zone will be added to the repository in an effort to improve visualization and transparency of the modelled area). Considering this approach and the wide range of saturation studied (from full saturation to < 1%), the catchment imports and exports water from neighboring catchments, but the buffer area guarantees a limited effect of the boundary on the catchments of interest. With that in mind, the purpose of the paper was not to trace recharge inside and outside the catchment but to observe the behavior of seepage area in the catchment. We will improve the description of the buffer zone and its impact on interbasin flow in section 2.3 detailing the numerical modelling approach.

2. To simplify the analysis, the authors assumed an aquifer thickness of 100 meters for all catchments with a homogeneous and isotropic hydraulic conductivity. Although these assumptions ease the comparison of these systems, there are some limitations that need to be addressed. For instance, what are the values of hydraulic conductivity used for each catchment? did the authors calculate an equivalent hydraulic conductivity for each catchment using the geology of each site? Also, the depth to bedrock may vary within each catchment depending on the geology; could these changes affect the seepage calculations? Lastly, it has been suggested that meteoric water can travel to depths of kilometers in areas with high topographic relief (i.e., Frisbee et al., 2017; McIntosh & Ferguson, 2021). Have the authors considered that some of the flowpaths that contribute to seepage or that export to other catchments could come from these deep groundwater systems? I suggest including these limitations within the manuscript to aid in discussing the results.

In this study, we do not aim to model the impact of specific-site geology and local heterogeneity. We only use the topographies of the Chilean landscapes for their diversity in landform shapes. It is a synthetic study that aims at extracting scaling laws between landforms, and groundwater seepage distribution. Thus, we did not assign site-specific K values nor calculate equivalent hydraulic conductivities based on local geology. Neither did we consider complex patterns of recharge. Instead, we investigate the impact of topographies over a wide range of values of the dimensionless ratio K/R, covering from humid to arid systems, from hard-rock to sedimentary environments. We may recall as well that analytical solutions to the diffusivity equation in unconfined conditions reveal that the piezometric level is directly controlled by the K/R ratio, indicating the pertinence of the K/R ratio for studying the surface-subsurface interaction and seepage area development.

We will revise the manuscript to clearly state these assumptions and discuss their implications, particularly in the context of extrapolating our findings to natural systems with complex geology and flow systems operating at multiple scales. We propose to include in Figure 3 the climatic-geological context along with the horizontal axis (K/R ratio).

3. Similarly, the authors assumed a uniform effective recharge for all the catchments. Did the authors assume probable recharge ranges for the area of study or arbitrarily pick recharge values to explore a wide range of K/R that presented desaturation? I recommend clarifying this part in the text.

As mentioned before, we chose to investigate a wide range of K/R to obtain the full range of possible saturation for each catchment consistently with various climatic contexts (humid, semi-arid, arid). Furthermore, to consider more realistic settings in terms of K/R, the fit on the desaturation function was weighted for seepage values < 20%. We will clarify that:

- The fit has been carried out on the possible saturation range in the text.
- The K/R ratio covers plausible climatic and geological contexts in Figure 3.
  - 4. This reviewer understands that this might be out of the scope of the study. However, the authors focused on the dimensionless ratio between hydraulic conductivity and recharge (K/R). This dimensionless quantity comes from previous studies done on 1D and 2D analysis (e.g., Bresciani et al., 2014). Have the authors considered doing a

dimensional analysis of their 3D model to explore what other dimensionless quantities arise from a catchment scale system? There might be other dimensionless quantities that relate aquifer thickness and drainage length, allowing another physical approach to relating the studied catchments.

Identifying additional scaling that integrates more fully the 3D nature of groundwater flow systems, and the influence of topographic metrics is interesting. Although we did not perform a full dimensional analysis in this study, we recognize the value of such an approach but it falls beyond the scope of this study. To the knowledge of the authors, there are no analytical solutions to support a 3D dimensional analysis. We will include a note in the discussion acknowledging this potential and the opportunity it presents for further research.

2. In line 157, within the results section, the authors state: "The fit shows minimal RMSE values between 0.01 and 0.08, indicating that seepage evolution with increasing *K*/*R* can be successfully parameterized with only two parameters,  $\lambda$  and *n*." These results come from the multiple assumptions made to the conceptual model. Have the authors considered if this conclusion will hold if the analyzed system is heterogeneous and anisotropic or if the thickness of the aquifer is variable within the system? Some of these thoughts can be addressed as possible limitations of the study.

Thank you for this comment. Following on the previous comments and answers, we will make sure to discuss more in depth the applications and limitations of our results in the discussion, with an emphasis on the potential impacts of heterogeneous and anisotropic cases.

3. By the end of the results section, starting in line 216, the authors briefly explain how they fitted a Random Forest model to predict the values of  $\lambda$  and *n* in other catchments based on PCA analysis of the topographic parameters. I suggest moving and expanding this explanation to the methods section, as it will aid in understanding the reasoning behind performing such an analysis.

**We agree - the detailed explanation about our Random Forest algorithm and study will be moved to the method sections.**

4. This analysis shows that some predicted catchments are clustered as lowland catchments (red contours) despite being located within the Andes Cordillera (Figure 4d at the northern part of the map). Are these correctly labeled? Are these there because they are located in flatter places within the Cordillera? Or are these outliers from the analysis? I suggest adding more information about these possible outliers.

These catchments are indeed well labeled as lowland catchments based on their landforms description (mostly flat), those catchments correspond to Altiplano, and endoreic catchments located in the Cordillera explaining their labelling. However, we agree with the reviewer that the term "lowland" in not appropriate in this case. Then, we propose to modify the cluster name from "lowlands" to "flat" to avoid confusion and will add detailed comments to the discussion section specifically on Altiplano flat catchments.

5. Based on the previous comments, I suggest including an additional paragraph or section that addresses the study's potential limitations. This addition can help describe the implications of these findings and the future work needed to address some of the assumptions.

We agree with the reviewer and will add and discuss potential limitations of our study.

**Technical Corrections**

Besides the comments described above, I have a few technical recommendations for the manuscript.

1. Line 27 states: "Considering steady-state groundwater flow systems, the depth of the water table, **and so the** distribution of flow paths..." Consider removing "so."

**The "so" will be removed.**

2. In lines 77 and 139, there is a reference to Figure 1a and Figure 1c, arguing that this figure shows the catchment colors and clusters. However, there is no reference to the colors and clusters in Figure 1. This might be missing from the figure, or the authors might reference Figure 4 instead. Please verify this inconsistency.

**The reference to the figure will be corrected.**

3. I suggest changing the equation numbering to "(1)" instead of "Equation 1."

The equations numbering will be modified.

---

## Author Comment (AC2)

**Our answers in *Italic*.**

**Reviewer #2 Evaluations:**

The authors presents an interesting method to predicts and estimated the seepage of wetland zone at varying the groundwater level and recharge rate on a wide variety of landformes in the region of Chile. The work is interesting and the scientific soundness is good. However there are several weakness. The introduction is poor and the objective of this study is not clear. Discussion is poor presenting a structure of a conclusion. In the attached file thera are other minor remarks.

The introduction will be improved clarifying the requested terms and further clarifying the objective of the study. Regarding the discussion, it will be further developed, especially considering reviewer 1 comments on work limitations.

1. Introduction: Please improve the introduction. It is appea a bit poor the the reader.

**The introduction will be improved.**

Line 28: "Recharge rate (R)": What do you intedn for recharge rate? is it the flow rate that leave the aquifer and recharge the wetlands? Please explain better this concept.

*The definition of the recharge rate will be clarified.*

Line 58: "m.a.s.l.": Please explain the acronym. It shoudl be amsl.

m.a.s.l. stands for "meters above sea level".

Line 140: "Figure 1a)": Figure 3?

*The reference to the corresponding figure will be corrected.*

Line 166: "(a) Normalized seepage area against [...]": Improve the quality of figure 1a

The quality of the figure will be improved and corrected.

4. Discussion and perspectives: Conclusions?

The discussion part will be extended.

---

## Author Response (AR1)

**Dear Associate Editor,**

Please find attached the revised version of our manuscript hess-2024-381 entitled *Catchment landforms predict groundwater-dependent wetland sensitivity to recharge changes* by Etienne Marti, Sarah Leray and Clément Roques. In the following document, we make a summary of the main improvements made to the paper, as well as detailed responses to reviewers' comments.

**In this document:**

- in black: editor and referee comments are shown
- in blue: responses to the referees with our relevant changes to the updated manuscript.

**Editor decision: Publish subject to minor revisions (further review by editor)**

Please, act according to the comments raised by the reviewers and reply to these comments to produce an acceptable version for publication in HESS.

We sincerely thank the editor for your appreciation of our work and the opportunity to address the reviewers' comments toward final acceptance in HESS.

**In the revised version of the manuscript:**

- The "lowlands" cluster was redefined to "flat" cluster to avoid possible confusion.
- We adapted the introduction section to consider comments of reviewer 2, improving the description and role of groundwater-dependent ecosystems. Furthermore, we clarified their importance in regard to climate change and how recharge changes could impact them.
- We reorganized the objectives of the study and added specific goals.
- In the methods section, we clarified modeling decisions especially on the buffer zone and its impact on interbasin flow.
- We added to the methods section a sub-paragraph to detail our Random Forest approach to predict our desaturation function parameters based on topographic assumptions.
- The discussion section is extended to consider the limitations of our study and possible future investigation questions arising from this study.
- A conclusion section was created to resume the work done on the paper.

**Reviewer #1 Evaluations:**

Thank you for submitting your Manuscript to HESS. Please find below the comments and suggestions made regarding the current version of the manuscript.

This manuscript aims to study the influence of topography on the desaturation rates of groundwater-dependent landscapes in response to changes in recharge. The authors achieve this objective by categorizing the catchments into lowland, transition, and mountain settings clusters, using geomorphon descriptors, and implementing 3D steady-state groundwater models to derive each catchment's flow partitioning and seepage area extent. The findings illustrate that mountains exhibit reduced seepage area compared to lowlands at equivalent hydraulic conductivity and recharge ratios but are less sensitive to fluctuations in recharge. Finally, the authors performed a correlation statistical analysis between the geomorphondefine landforms and the desaturation indicators, which, according to them, enables the prediction of sensitivity to climate change based on topographic analysis.

I have read the manuscript with great interest. My overall opinion is that the manuscript is well-written but needs some moderate revisions and clarifications. Below, I have listed comments, hoping they may help improve the manuscript's quality.

We would like to thank the reviewer for the interest shown in our work and for the constructive evaluation. We appreciate the recognition of our modelling approach, and the valuable suggestions provided to improve clarity. All comments will be carefully addressed and incorporated into the revised manuscript.

**Specific Comments**

- 1. Some clarification is needed regarding the 3D numerical model approach.
  - 1. The authors extended the model domain by 20% of the total area of the catchment to avoid boundary effects. In the extended domain, did the authors use the elevation of adjacent catchments? Have the authors considered possible interbasin flow happening among catchments, as described in Fan (2019)? If so, are there any implications for this interbasin flow in the general results for lowland settings?

Indeed, the elevation of the adjacent catchments was used. Recharge was applied on all model cells, including the buffer area around the catchment, while the catchment is in the center of the simulated area. The 60 modeled catchments and their boundaries were validated by analyzing the topographic context in the area (topographic maps of each catchment and its buffer zone will be added to the repository in an effort to improve visualization and transparency of the modelled area). Considering this approach and the wide range of saturation studied (from full saturation to < 1%), the catchment imports and exports water from neighboring catchments, but the buffer area guarantees a limited effect of the boundary on the catchments of interest. With that in mind, the purpose of the paper was not to trace recharge inside and outside the catchment but to observe the behavior of seepage area in the catchment.

To reflect these changes, we re-organize and clarify the methodology as follow: L.118-127

"To limit boundary effects, a buffer zone extending the model domain by 20% around each catchment was added. A sensitivity analysis (Abhervé et al., 2023) of the extent of the buffer zone was performed to ensure that no impact on the seepage distribution was identified within the studied catchment (Figure 2a). The model bottom mirrored the topography with a 100m-thick aquifer (Figure 2b). Assuming a constant aquifer thickness minimized the potential effects of transmissivity changes on seepage distribution. The 100m thickness was chosen to realistically accommodate both flat sedimentary catchments and steep mountainous aquifers (Condon et al., 2020). The side and bottom boundaries of the buffer box were set as no-flow. For generality, effective recharge *R* was uniformly set at the water table across both the catchment and its buffer, enabling the simulation of both inflow and outflow across the model boundaries. This setup allowed considering interbasin groundwater exchanges which are particularly likely to occur under low water table conditions (Fan, 2019)."

1. To simplify the analysis, the authors assumed an aquifer thickness of 100 meters for all catchments with a homogeneous and isotropic hydraulic conductivity. Although these assumptions ease the comparison of these systems, there are some limitations that need to be addressed. For instance, what are the values of hydraulic conductivity used for each catchment? did the authors calculate an equivalent hydraulic conductivity for each catchment using the geology of each site? Also, the depth to bedrock may vary within each catchment depending on the geology; could these changes affect the seepage calculations? Lastly, it has been suggested that meteoric water can travel to depths of kilometers in areas with high topographic relief (i.e., Frisbee et al., 2017; McIntosh & Ferguson, 2021). Have the authors considered that some of the flowpaths that contribute to seepage or that export to other catchments could come from these deep groundwater systems? I suggest including these limitations within the manuscript to aid in discussing the results.

In this study, we do not aim to model the impact of specific-site geology and local heterogeneity. We only use the topographies of the Chilean landscapes for their diversity in landform shapes. It is a synthetic study that aims at extracting scaling laws between landforms, and groundwater seepage distribution. Thus, we did not assign site-specific K values nor calculate equivalent hydraulic conductivities based on local geology. Neither did we consider complex patterns of recharge. Instead, we investigate the impact of topographies over a wide range of values of the dimensionless ratio K/R, covering from humid to arid systems, from hard-rock to sedimentary environments. We may recall as well that analytical solutions to the diffusivity equation in unconfined conditions reveal that the piezometric level is directly controlled by the K/R ratio, indicating the pertinence of the K/R ratio for studying the surface-subsurface interaction and seepage area development.

We revised the manuscript to clearly state these assumptions and discuss their implications as follow L298-309:

"While the aim of the present work is to establish a comprehensive exploration of landform controls on seepage dynamics, several simplifications limit its direct application to specific real-catchment systems. Although the models are based on real topographies from the Chilean Andes, the experiment does not intend to capture actual complexity of hydrogeological systems, but rather to explore a wide enough range of natural landform geometries for comparative analysis. First, we assumed homogeneous and isotropic aquifer properties with a fixed aquifer thickness, thereby neglecting geological heterogeneities, anisotropy, and variability in the depth of the active groundwater flow system (Frisbee et al., 2017; McIntosh & Ferguson, 2021), and consequently the seepage distribution, that can be involved in real landscape. While our use of the dimensionless K/R ratio offers a robust approach for analyzing desaturation responses, future research could benefit from exploring additional parameters that account for catchment geometry, relief, or flow system depth. Additionally, the model results presented here operate under steady-state conditions and exclude the potential impacts of seasonal recharge variability, vegetation feedbacks, or the

role of the unsaturated zone near the land surface. Exploring such processes, especially under transient conditions and with heterogeneous parameters, represents a promising perspective for future research."

1. Similarly, the authors assumed a uniform effective recharge for all the catchments. Did the authors assume probable recharge ranges for the area of study or arbitrarily pick recharge values to explore a wide range of K/R that presented desaturation? I recommend clarifying this part in the text.

As mentioned before, we chose to investigate a wide range of K/R to obtain the full range of possible saturation for each catchment consistently with various climatic contexts (humid, semi-arid, arid). Furthermore, to consider more realistic settings in terms of K/R, the fit on the desaturation function was weighted for seepage values < 20%. We clarified that the fit has been carried out on the possible saturation range in the text (L148).

1. This reviewer understands that this might be out of the scope of the study. However, the authors focused on the dimensionless ratio between hydraulic conductivity and recharge (*K/R*). This dimensionless quantity comes from previous studies done on 1D and 2D analysis (e.g., Bresciani et al., 2014). Have the authors considered doing a dimensional analysis of their 3D model to explore what other dimensionless quantities arise from a catchment scale system? There might be other dimensionless quantities that relate aquifer thickness and drainage length, allowing another physical approach to relating the studied catchments.

Identifying additional scaling that integrates more fully the 3D nature of groundwater flow systems, and the influence of topographic metrics is interesting. Although we did not perform a full dimensional analysis in this study, we recognize the value of such an approach but it falls beyond the scope of this study. To the knowledge of the authors, there are no analytical solutions to support a 3D dimensional analysis. We included a note in the discussion acknowledging this potential and the opportunity it presents for further research in L304-306:

"While our use of the dimensionless K/R ratio offers a robust approach for analyzing desaturation responses, future research could benefit from exploring additional parameters that account for catchment geometry, relief, or flow system depth."

1. In line 157, within the results section, the authors state: "The fit shows minimal RMSE values between 0.01 and 0.08, indicating that seepage evolution with increasing K/R can be successfully parameterized with only two parameters,  $\lambda$  and n." These results come from the multiple assumptions made to the conceptual model. Have the authors considered if this conclusion will hold if the analyzed system is heterogeneous and anisotropic or if the thickness of the aquifer is variable within the system? Some of these thoughts can be addressed as possible limitations of the study.

Thank you for this comment. Following on the previous comments and answers, we made sure to discuss more in depth the applications and limitations of our results in the discussion, with an emphasis on the potential impacts of heterogeneous and anisotropic cases (L298-309).

1. By the end of the results section, starting in line 216, the authors briefly explain how they fitted a Random Forest model to predict the values of  $\lambda$  and n in other catchments based on PCA analysis of the topographic parameters. I suggest moving and expanding this explanation to the methods section, as it will aid in understanding the reasoning behind performing such an analysis.

We agree - the detailed explanation about our Random Forest algorithm and study has been moved to the method sections (section 2.4. Regionalization with Random Forest Algorithm) and consider the following explanation (L155-167):

"To predict the desaturation response metrics  $\lambda$  and n from topographic descriptors and regionalize our findings, we employed Random Forest regression using the scikit-learn library in Python. The input features for the model were the first two principal components (PC1 and PC2) derived from the sixty catchments. Random Forest models were trained independently for  $\lambda$  and n. Model performance and robustness were assessed using a bootstrap resampling procedure with 5,000 iterations. In each iteration, 10 catchments were randomly selected as a test set, while the remaining 50 were used for training. The coefficient of determination (R²) was calculated on the test data for each iteration, and the resulting R² distribution was used to evaluate model reliability (see Supplementary Material S2 for Kernel Density Estimate of R² values). Hyperparameter tuning for each model was performed using GridSearchCV with 2-fold cross-validation within each training subset. The tested parameter grid included n\_estimators  $\in$  {50, 200, 500, 1000} and max\_depth  $\in$  {None, 2, 10, 20}. The best combination of hyperparameters was used to retrain the model on the full training set in each iteration. The final Random Forest model was defined as the one achieving the best trade-off in predictive accuracy for both  $\lambda$  and n, and it was applied to predict desaturation metrics in sixty-three additional catchments located in South Chile."

1. This analysis shows that some predicted catchments are clustered as lowland catchments (red contours) despite being located within the Andes Cordillera (Figure 4d at the northern part of the map). Are these correctly labeled? Are these there because they are located in flatter places within the Cordillera? Or are these outliers from the analysis? I suggest adding more information about these possible outliers.

These catchments are indeed well labeled as lowland catchments based on their landforms description (mostly flat), those catchments correspond to Altiplano, and endoreic catchments located in the Cordillera explaining their labelling. However, we agree with the reviewer that the term "lowland" in not appropriate in this case. Then, we modified the cluster name from "lowlands" to "flat" to avoid confusion and we added the following comment to the discussion section specifically on Altiplano flat catchments (L274-277):

"Moreover, the proposed methodology demonstrated strong robustness to outliers and atypical landscape configurations. For example, the Andes Mountains in northern Chile include the "Altiplano" region—characterized by extensive flat areas within an otherwise mountainous setting. The method successfully identified such catchments and classified them

as flat (Figure 4d between 19 and 23°S), illustrating its capacity to perform reliably across diverse geomorphological contexts."

1. Based on the previous comments, I suggest including an additional paragraph or section that addresses the study's potential limitations. This addition can help describe the implications of these findings and the future work needed to address some of the assumptions.

We agree with the reviewer and added and discussed more thoroughly potential limitations of our study (L298-309).

**Technical Corrections**

Besides the comments described above, I have a few technical recommendations for the manuscript.

1. Line 27 states: "Considering steady-state groundwater flow systems, the depth of the water table, **and so the** distribution of flow paths..." Consider removing "so."

**The "so" was removed.**

1. In lines 77 and 139, there is a reference to Figure 1a and Figure 1c, arguing that this figure shows the catchment colors and clusters. However, there is no reference to the colors and clusters in Figure 1. This might be missing from the figure, or the authors might reference Figure 4 instead. Please verify this inconsistency.

The references to the figures were corrected.

1. I suggest changing the equation numbering to "(1)" instead of "Equation 1."

The equations numbering was modified.

**Reviewer #2 Evaluations:**

The authors presents an interesting method to predicts and estimated the seepage of wetland zone at varying the groundwater level and recharge rate on a wide variety of landformes in the region of Chile. The work is interesting and the scientific soundness is good. However there are several weakness. The introduction is poor and the objective of this study is not clear. Discussion is poor presenting a structure of a conclusion. In the attached file there are other minor remarks.

The introduction has been improved further clarifying the objective of the study. Regarding the discussion, it has extended, especially considering reviewer 1 comments on work limitations. Regarding the objectives of the paper, they were modified as (L59-63):

"Additionally, we aim to identify appropriate topographic indicators that explain the variety of hydrogeological responses and provide statistical means to extrapolate our findings to ungauged or data-scarce regions. By linking geomorphologic patterns to seepage dynamics, we seek to improve the prediction of wetland desaturation risk under changing climate conditions. This approach supports the development of transferable frameworks for assessing groundwater-dependent ecosystem vulnerability across heterogeneous terrain."

1. Introduction: Please improve the introduction. It is appea a bit poor the the reader.

The introduction has been improved, including amor thorough description of the role and importance of groundwater dependent ecosystems (L24-35):

"Changes in precipitation regimes and increasing temperatures driven by climate change are anticipated to significantly affect both surface and subsurface water resources (Berghuijs et al., 2024; Taylor et al., 2013; Konapala et al., 2020). Extended drought periods and reduced recharge are expected to threaten the functioning of groundwater-dependent ecosystems (Rohde et al., 2024; Tetzlaff et al., 2024). These ecosystems rely on groundwater contributions, to maintain their ecological structure and functional integrity, including processes that support biodiversity and key ecosystem services (Eamus et al., 2006; Barron et al., 2014; Doody et al., 2017). They encompass both terrestrial and aquatic environments, including wetlands, springs, rivers (riparian, aquatic, and hyporheic zones), lakes, grasslands, forests, as well as coastal and estuarine habitats (Eamus et al., 2006; Kløve et al., 2011). The extent to which groundwater-dependent ecosystems are vulnerable to climate-induced reductions in recharge depends not only on the hydrogeological properties of the underlying aquifer, but also on the role of landscape morphology in shaping groundwater flow and discharge patterns (Gleeson & Manning, 2008; Singha & Navarre-Sitchler, 2022). Identifying the physical controls on groundwater emergence at the land surface is therefore essential to improve our ability to anticipate groundwater-dependent ecosystems responses to climate variability."

Line 28: "Recharge rate (R)": What do you intedn for recharge rate? is it the flow rate that leave the aquifer and recharge the wetlands? Please explain better this concept.

The definition of the recharge rate has been specified as "groundwater recharge rate".

Line 58: "m.a.s.l.": Please explain the acronym. It should be amsl.

m.a.s.l. stands for "meters above sea level".

Line 140: "Figure 1a)": Figure 3?

The reference to the corresponding figure has been corrected.

Line 166: "(a) Normalized seepage area against [...]": Improve the quality of figure 1a

The quality of the figure has been improved and the high-resolution figure files (.pdf) are attached with the revised manuscript.

1. Discussion and perspectives: Conclusions?

The discussion part has been extended to consider limitations of the study as mentioned for reviewer 1 (L298-309) and a separated conclusion section was introduced. (L310-319):

"To conclude, our study demonstrates that catchment-scale topographic features, quantified through geomorphon-based landform classification, exert a first-order control on groundwater seepage dynamics under varying recharge conditions. By linking these landform metrics to a desaturation function, we show that the sensitivity of groundwater seepage extent to climate variability can be predicted from topography alone. This insight enables the development of a robust and scalable framework for assessing hydroclimatic vulnerability, particularly relevant for data-scarce regions. The ability to regionalize desaturation behavior using simple statistical learning tools, such as Random Forests as presented here, opens up new opportunities for applying this approach to ungauged basins in other regions (Hrachowitz et al., 2013). As such, our findings offer not only a methodological advance, but also enable potential for its application to assess the vulnerability of regional scale groundwater-dependent wetlands and the ecosystem they support to climate change."